# Impact of Energy and Protein Delivery to Critically Ill Patients: A Systematic Review and Meta-Analysis of Randomized Controlled Trials

**DOI:** 10.3390/nu14224849

**Published:** 2022-11-16

**Authors:** Nobuto Nakanishi, Shinya Matsushima, Junko Tatsuno, Keibun Liu, Takahiko Tamura, Hiroshi Yonekura, Norimasa Yamamoto, Takeshi Unoki, Yutaka Kondo, Kensuke Nakamura

**Affiliations:** 1Division of Disaster and Emergency Medicine, Department of Surgery Related, Kobe University Graduate School of Medicine, Kobe 650-0017, Japan; 2Department of Rehabilitation Center, St. Marianna University School of Medicine Hospital, Kawasaki 216-8511, Japan; 3Department of Quality Management, Kokura Memorial Hospital, Kitakyusyu 802-8555, Japan; 4Critical Care Research Group, The Prince Charles Hospital, Brisbane, QLD 4032, Australia; 5Faculty of Medicine, The University of Queensland, Brisbane, QLD 4006, Australia; 6Department of Anesthesiology and Intensive Care Medicine, Kochi Medical School, Nankoku City 783-8505, Japan; 7Department of Anesthesiology and Pain Medicine, Fujita Health University Bantane Hospital, Nagoya City 454-8509, Japan; 8Department of Acute and Critical Care Nursing, Toyama Chuo Hospital, Toyama City 930-8550, Japan; 9Department of Acute and Critical Care Nursing, School of Nursing, Sapporo City University, Sapporo 005-0864, Japan; 10Department of Emergency and Critical Care Medicine, Juntendo University Urayasu Hospital, Urayasu 279-0021, Japan; 11Department of Emergency Medicine, Teikyo University Hospital, Tokyo 173-8606, Japan

**Keywords:** critically ill patient, nutrition therapy, energy, protein, physical impairment

## Abstract

Optimal energy and protein delivery goals for critically ill patients remain unknown. The purpose of this systematic review and meta-analysis was to compare the impact of energy and protein delivery during the first 4 to 10 days of an ICU stay on physical impairments. We performed a systematic literature search of MEDLINE, CENTRAL, and ICHUSHI to identify randomized controlled trials (RCTs) that compared energy delivery at a cut-off of 20 kcal/kg/day or 70% of estimated energy expenditure or protein delivery at 1 g/kg/day achieved within 4 to 10 days after admission to the ICU. The primary outcome was activities of daily living (ADL). Secondary outcomes were physical functions, changes in muscle mass, quality of life, mortality, length of hospital stay, and adverse events. Fifteen RCTs on energy delivery and 14 on protein were included in the analysis. No significant differences were observed in any of the outcomes included for energy delivery. However, regarding protein delivery, there was a slight improvement in ADL (odds ratio 21.55, 95% confidence interval (CI) −1.30 to 44.40, *p* = 0.06) and significantly attenuated muscle loss (mean difference 0.47, 95% CI 0.24 to 0.71, *p* < 0.0001). Limited numbers of RCTs were available to analyze the effects of physical impairments. In contrast to energy delivery, protein delivery ≥1 g/kg/day achieved within 4 to 10 days after admission to the ICU significantly attenuated muscle loss and slightly improved ADL in critically ill patients. Further RCTs are needed to investigate their effects on physical impairments.

## 1. Introduction

Critically ill patients experience significant skeletal muscle atrophy and physical impairments [1]. Consequently, one third of critically ill patients have prolonged physical impairments, known as post intensive care syndrome [2]. Therefore, it is important to provide adequate nutrition therapy as a preventative strategy [3]. However, optimal energy and protein delivery goals in the acute phase to improve physical impairments in critically ill patients currently remain unclear.

Previous systematic reviews and meta-analyses have mostly been conducted based on mortality and infection, not physical impairments. The majority of nutritional guidelines recommend lower energy delivery than expenditure in the acute phase [3,4,5,6], called permissive underfeeding, based on previous findings showing a lower incidence of infection [7]. The American Society for Parenteral and Enteral Nutrition (ASPEN) recommends achieving 12–25 kcal/kg in 7–10 days after admission to the ICU, while the European Society for Clinical Nutrition and Metabolism (ESPEN) recommends 70% of estimated energy expenditure during the 1st week of an ICU stay in cases with no access to indirect calorimetry. However, the effects of these delivery goals on physical impairments are unknown.

An optimal protein delivery goal has not yet been established. Most guidelines recommend protein delivery >1 g/kg/day [3,4,6]; however, this is based on weak evidence. Furthermore, recent guidelines recommend protein delivery <1 g/kg/day for a shorter hospital stay [5]. These disparities may be attributed to outcome settings. It may be more important to set the outcome of nutrition therapy to physical impairments rather than mortality or the hospital stay.

Another reason for this discrepancy may be the timing of an intervention in the acute phase, namely, the early or late period. In the early period of the acute phase, a consensus was reached on early enteral nutrition and permissive underfeeding [3]. However, the late period of the acute phase may require sufficient energy and protein delivery due to necessary expenditure. Although the acute phase consists of the early and late periods, most guidelines have summarized the acute phase without these two periods and conducted a systematic review. Therefore, the issue regarding the timing of interventions may lead to discrepancies among recommendations. Further systematic reviews are needed to exclude the early period of the acute phase and focus on the late period, namely the first 4 to 10 days of an ICU stay.

Since some randomized controlled trials (RCTs) have investigated the effects of nutrition therapy on physical impairments, further systematic reviews and meta-analyses that include these RCTs are required. Therefore, we herein reviewed current evidence to assess the impact of energy and protein delivery during the first 4 to 10 days of an ICU stay on physical impairments. According to previous studies and guidelines, the cut-off value for energy delivery was set to 20 kcal/kg/day or 70% of estimated energy expenditure [3] and that for protein delivery was 1 g/kg/day [5].

## 2. Materials and Methods

### 2.1. Protocols and Registration

This systematic review protocol has been registered in the International Prospective Register of Systematic Reviews (PROSPERO, CRD42021285548) [8]. The protocol adhered to the Preferred Reporting Items for Systematic Reviews and Meta-analyses (PRISMA) statement and checklist [9]. The final report was based on PRISMA guidelines.

### 2.2. Search Strategies

We systematically searched for articles published in MEDLINE via PubMed (until 2 November 2021), the Cochrane Central Register of Controlled Trials (CENTRAL, until 3 November 2021), and the Igaku Chuo Zasshi database (ICHUSHI, until 2 November 2021). Searches were not restricted by language, the publication status, date of publication, or sample size. A manual search was also conducted to identify all potentially relevant articles until 3 November 2021. We listed key search terms in the Appendix A.

### 2.3. Study Selection

Two authors (RK, HY for energy and NN, TT for protein) conducted the comprehensive first-line literature search for clinical trials conducted on humans. The results obtained were exported into Mendeley Desktop (Version 1.19.8) for screening and the removal of duplicates. After duplicate removal, the authors (RK, HY for energy and NN, TT for protein) independently screened study titles and abstracts for potential relevance. When a disagreement was identified between reviewers, the full text of the paper was retrieved, and the disagreement was discussed until a consensus was reached. The full text of articles included in the final selection was independently reviewed by authors (RK, HY for energy and NN, TT for protein).

### 2.4. Eligibility Criteria

Study types: RCTs were included, whereas non-randomized and observational studies were excluded from this meta-analysis.

Population: The included population was critically ill adult patients (≥18 years of age) admitted to the ICU. No specific diagnostic criteria were used for population selection.

Intervention: The intervention was (1) energy delivery ≥20 kcal/kg/day or ≥70% of estimated energy expenditure achieved within 4 to 10 days after admission to the ICU, or (2) protein delivery ≥1 g/kg/day achieved within 4 to 10 days after admission to the ICU.

Control: The control group received (1) energy delivery <20 kcal/kg/day or 70% of estimated energy expenditure achieved within 4 to 10 days after admission to the ICU, or (2) protein delivery <1 g/kg/day achieved within 4 to 10 days after admission to the ICU.

Outcome: The primary outcome was activities of daily living (ADL), evaluated as the Barthel Index (BI) [10] and/or Functional Independence Measure (FIM) [11] at the time of hospital discharge or in the follow-up period for up to 1 year. Secondary outcomes were as follows: (1) muscle strength and physical functions, including handgrip strength, Medical Research Council-sum scores [12], Short Physical Performance Battery [13], and the 6-min walk distance [14] at the time of hospital discharge or in the follow-up period for up to 1 year; (2) Changes in muscle mass during hospitalization, which were assessed by ultrasound or computed tomography [15], not anthropometrics due to insufficient reliability [16]. Ultrasound and computed tomography have good inter-class correlation at 0.97 and 0.95–0.99, respectively [16,17]; (3) Quality of life (QOL) scores at the time of hospital discharge or in the follow-up period for up to 1 year, including the Short Form Health Survey (SF-36/8) [18,19], RAND-36 [20], and EuroQol 5 dimensions (EQ-5D) [21,22]; (4) Mortality (hospital mortality or 28- or 90-day mortality, whichever was longer); (5) length of hospital stay, and (6) all adverse events related to nutrition therapy.

### 2.5. Risk of Bias Assessment

The risk of bias was assessed according to the “Cochrane Handbook for Systematic Reviews of Interventions” [23]. The following risk of bias components was evaluated by two authors (RK, HY for energy and NN, TT for protein): sequence generation, allocation concealment, blinding of participants, blinding of investigators, incomplete outcome data, selective outcome reporting, and other biases, including the source of funding bias. Disagreements between two authors were resolved by a discussion or the adjudication of another author (JT for energy and SM for protein). The risk of bias was considered to be high when bias was likely to affect outcomes and low when it was not present, or present but unlikely to affect outcomes. The risk of bias was considered to be unclear if there was insufficient information to classify each element as high or low.

### 2.6. Grading of Recommendations Assessment, Development and Evaluation Approach

Two authors (RK, HY for energy and NN, TT for protein) independently assessed the quality of evidence (QoE) for outcomes using the Grading of Recommendations Assessment, Development, and Evaluation (GRADE) tool [24]. Although QoE is a continuum, we assessed it for each outcome categorized as high, moderate, low, or very low using the GRADEpro Guideline Development Tool [25].

### 2.7. Data Extraction

The title, authors, year of publication, journal name, and abstract of each included article were identified. Conference abstracts were excluded from the search results. Two review authors independently extracted data related to the study population, inclusion criteria, exclusion criteria, sample size, interventions, comparisons, potential biases in the conduct of the trial, and outcomes including adverse events from the original reports into a spreadsheet.

### 2.8. Data Summarization, Heterogeneity, and Synthesis

We performed a meta-analysis according to the “Cochrane Handbook for Systematic Reviews of Interventions” and PRISMA (Preferred Reporting Items for Systematic Reviews and Meta-Analyses) guidelines using the software Review Manager (RevMan 5.3, Copenhagen, Denmark: The Nordic Cochrane Centre, the Cochrane Collaboration 2014). We pooled estimates using a random effects model. The risk ratio was estimated for dichotomized outcomes and the mean difference or standard mean difference for continuous outcomes. To assess between-study heterogeneity, the Cochran Q statistic was calculated and I-squared was used to quantify the magnitude of between-study heterogeneity. If significant heterogeneity was found, the median of estimates was reported rather than a weighted, pooled estimate. Random-effect meta-analyses were conducted to estimate the pooled risk ratio (dichotomized outcomes) or mean difference (continuous outcomes).

## 3. Results

### 3.1. Literature Search

The article selection process is shown in the PRISMA Flow Diagram in Figure 1. The search strategy identified 1420 records for energy and 1419 for protein, of which 49 and 121, respectively, were judged to be potentially eligible based on the abstract. After the exclusion of 30 and 105 of these records following a full-text review, 15 RCTs for energy (Table 1) [26,27,28,29,30,31,32,33,34,35,36,37,38,39,40,41,42,43] and 14 for protein (Table 2) [27,44,45,46,47,48,49,50,51,52,53,54,55,56] were included in the analysis.

### 3.2. Characteristics of Included Studies

Table 1 and Table 2 show a summary of study characteristics. Numbers, ages, and characteristics are shown as well as the nutrition delivery route. Intervention settings and the period are also summarized in these tables. The setting and delivery route varied among protocols. We did not contact the authors to acquire additional information.

### 3.3. Risk of Bias in Included Studies

The risk of bias in the included studies is shown in Appendix A. Regarding blinding of participants and personnel, 14/15 RCTs for energy and 7/14 for protein had a high risk of bias. The blinding of assessments was not described in all studies. An attribution bias was observed in many studies because of the difficulties associated with assessing physical function and patients who dropped out of the follow-up.

### 3.4. Outcomes

All outcomes were assessed and are summarized in the Appendix A. The GRADE working group grades of evidence were used as follows. High quality: further research is very unlikely to change our confidence in the estimate of the effect. Moderate quality: further research is likely to have an important impact on our confidence in the estimate of the effect and may change the estimate. Low quality: further research is very likely to have an important impact on our confidence in the estimate of the effect and is likely to change the estimate. Very low quality: we are very uncertain about the estimate.

### 3.5. Primary and Secondary Outcomes of Energy Delivery

#### 3.5.1. ADL

No previous RCT assessed BI or FIM for the optimal energy delivery dose.

#### 3.5.2. Physical Functions

Two RCTs reported effects on handgrip strength (Figure 2). Needham et al. and Ridley et al. evaluated handgrip strength at 12 months after acute lung injury and hospital discharge, respectively [27,33]. No significant increase was observed in this parameter with high energy delivery (2 trials, *n* = 192; MD 0.58, 95% CI −4.77 to 5.92, *p* = 0.83; I^2^ = 0%). The risk of bias was considered to be serious, and imprecision was very serious because of the small number of patients included. These serious risks led to the downgrading of QoE to very low.

#### 3.5.3. Changes in Muscle Mass

No previous study assessed the effects of the optimal energy delivery dose on changes in muscle mass.

#### 3.5.4. QOL Scores

Two RCTs reported effects on EQ-5D within one year of hospital discharge as a QOL score. We showed the result of evaluations at 12 months after acute lung injury by Needham et al. and at hospital discharge by Ridley et al. [27,33]. No significant increase was observed in this parameter with high energy delivery (2 trials, *n* = 551; MD 0.01, 95% CI −0.03 to 0.05, *p* = 0.74). The risk of bias was considered to be very serious, and imprecision was also very serious because of the small number of patients included. These serious risks led to the downgrading of QoE to very low.

#### 3.5.5. Mortality

Eight RCTs reported effects on mortality. No significant increase was observed in this parameter with high energy delivery (8 trials, *n* = 2754; MD 1.00, 95% CI 0.88 to 1.14, *p* = 0.95; I^2^ = 0%). QoE was graded as moderate.

#### 3.5.6. Length of Hospital Stay

Six RCTs reported effects on the length of hospital stay. No significant increase was noted in this parameter with high energy delivery (6 trials, *n* = 734; MD −1.08, 95% CI −4.86 to 2.70, *p* = 0.58; I^2^ = 45%). QoE was graded as moderate.

#### 3.5.7. Adverse Events

The reported adverse events of energy delivery were diarrhea, a residual volume in the stomach >300 mL, abdominal distention, vomiting, hyperglycemia, hypoglycemia, and gastrointestinal intolerance. No significant differences were observed in these adverse events with high energy delivery.

A forest plot summarized the impact of energy delivery during the first 4 to 10 days of an ICU stay on physical impairments. No significant differences were observed in the included outcomes.

### 3.6. Primary and Secondary Outcomes of Protein Delivery

#### 3.6.1. ADL

Three studies assessed BI at the hospital discharge on optimal protein delivery doses. No significant improvements were noted in ADL with protein delivery ≥1 g/kg/day (3 trials, *n* = 236; OR 21.55, 95% CI −1.30 to 44.40, *p* = 0.06; I^2^ = 76%). The risk of bias was not considered to be serious. Inconsistency and imprecision were serious because of strong heterogeneity and the small sample size, respectively. Therefore, QoE was downgraded to low because of these serious risks (Figure 3).

#### 3.6.2. Physical Functions

Two RCTs reported effects on handgrip strength. Fetterplace et al. and Ridley et al. evaluated the handgrip strength at the ICU discharge and hospital discharge, respectively [27,49]. No significant increase was observed in this parameter with high protein delivery (2 trials, *n* = 65; MD −1.00, 95% CI −5.79 to 3.79, *p* = 0.68; I^2^ = 0%). The risk of bias was considered to be serious, and imprecision was very serious because of the small number of patients included. These serious risks led to the downgrading of QoE to very low.

#### 3.6.3. Changes in Muscle Mass

Three RCTs reported effects on muscle mass changes. One study evaluated muscle mass by ultrasound, and two by computed tomography [45,53,55]. A significant increase was noted in this parameter with high protein delivery (three trials, *n* = 286; MD 0.47, 95% CI 0.24 to 0.71, *p* < 0.0001; I^2^ = 0%). Except for imprecision, other certainty assessment components were not serious. Therefore, certainty was graded as moderate.

#### 3.6.4. QOL Scores

Four RCTs reported effects on EQ-5D or RAND-36 within one year of hospital discharge as a QOL score. We showed the result of evaluations at the hospital discharge by Nakamura et al. and 90 days after the study by Ridley et al., Doig et al., Zhu et al. [27,53,54,56]. No significant increase was observed in this parameter with high protein delivery (four trials, *n* = 25; MD −0.10, 95% CI −0.24 to 0.03, *p* = 0.14; I^2^ = 0%). The risk of bias was considered to be serious, and imprecision was also serious because of the small number of patients included. These serious risks led to the downgrading of QoE to low.

#### 3.6.5. Mortality

Eleven RCTs reported effects on mortality. No significant increase was observed in this parameter with high protein delivery (11 trials, *n* = 1528; MD 0.90, 95% CI 0.73 to 1.12, *p* = 0.34; I^2^ = 0%). QoE was graded as moderate.

#### 3.6.6. Length of Hospital Stay

Thirteen RCTs reported effects on the length of hospital stay. No significant increase was noted in this parameter with high protein delivery (13 trials, *n* = 1644; MD 0.36, 95% CI −0.98 to 1.70, *p* = 0.60; I^2^ = 61%). QoE was graded as very low.

#### 3.6.7. Adverse Events

The reported adverse events of protein delivery were diarrhea, vomiting, a high gastric residual volume, and infection. No significant differences were observed in these adverse events with high protein delivery.

A forest plot summarized the impact of protein delivery during the first 4 to 10 days of an ICU stay on physical impairments. There was a slight improvement in ADL (OR 21.55, 95% CI −1.30 to 44.40, *p* = 0.06) and the attenuation of muscle loss (MD 0.47, 95% CI 0.24 to 0.71, *p* < 0.0001).

## 4. Discussion

In this systematic review and meta-analysis of optimal energy or protein delivery for critically ill patients, we found that high energy or protein delivery did not significantly affect most physical impairments, including QOL scores, mortality, the length of hospital stay, and adverse events. However, protein delivery >1 g/kg/day was associated with the attenuation of muscle loss and slight improvements in ADL (*p* = 0.06). Further studies on long-term physical impairments are needed because fewer studies set outcomes on ADL, physical functions, and QOL scores than mortality and the length of hospital stay.

To the best of our knowledge, this is the first systematic review and meta-analysis with two aspects. It focused on patient-centered outcomes rather than mortality and the length of hospital stay. Therefore, our outcome settings were mainly ADL, muscle mass changes, physical function, and QOL. The majority of systematic reviews previously focused on mortality and the length of hospital stay [4,5,57]; however, prolonged physical impairments are considered to be more important outcomes for preventing post intensive care syndrome. Through this systematic review, we identified a few studies that set the study outcome to physical impairments, requiring the careful consideration of outcome settings. We also focused on the late period of the acute phase during the first 4 to 10 days of an ICU stay. Most guidelines conducted systematic reviews on the acute phase, such as one week, without dividing it into the early or late period [3,4,5]. Although underfeeding in the early acute phase may be permitted, the late acute phase may require high energy and protein delivery to prevent physical impairments. Our systematic review is based on the late period of the acute phase.

One important result of the present study is that protein delivery >1 g/kg/day correlated with the prevention of muscle atrophy. The results obtained on muscle mass were consistent with a previous meta-analysis showing that higher protein delivery was associated with the attenuation of muscle loss [57]. Lee et al. reported that a 0.46 g/kg/day higher protein delivery was associated with the attenuation of muscle loss of 3.4% per week [57]. This meta-analysis included five RCTs, some of which used insufficient muscle mass assessment methods, such as arm circumference [49]. In critically ill patients, arm circumference is easily influenced by the fluid balance [16]. Our meta-analysis included studies that used ultrasound or computed tomography for muscle mass assessments, and we found that higher protein delivery was associated with the prevention of muscle atrophy. Although adverse events may occur with high protein delivery, no significant adverse events were detected with high protein delivery during the first 4 to 10 days of an ICU stay [58].

Protein delivery >1 g/kg/day was associated with long-term ADL assessed by the BI (*p* = 0.06). Few studies have investigated the relationship between nutritional therapy and long-term ADL. The malnutrition status at admission is generally associated with impaired ADL at hospital discharge or thereafter [59,60], and this has also been reported for critically ill patients [61]. Therefore, the prevention of malnutrition by nutrition therapy may improve long-term ADL because early rehabilitation has been shown to improve long-term ADL [62]. Since ESPEN recommends nutrition therapy coupled with physical therapy [3], high protein delivery may contribute to long-term ADL with sufficient rehabilitation.

High energy delivery had few effects on handgrip strength and QOL scores. As a result, the beneficial effects of high energy delivery were limited; however, there did not appear to be any adverse effects. Although evidence was generally very low, energy delivery >20 kcal/kg/day or 70% of estimated energy expenditure achieved within 4 to 10 days after admission to the ICU is plausible because energy is essential for rehabilitation and mobilization. Although most guidelines recommend permissive underfeeding, the optimal timing to deliver necessary energy has not yet been established. The ASPEN guidelines recommend energy delivery of 12–25 kcal/kg/day in the 7–10 days after admission to the ICU, whereas the ESPEN guidelines recommend <70% of required energy during the first two days to 80–100% three days after admission to the ICU. High energy delivery is safe and not a severe burden on medical staff; however, adverse events, such as parenteral nutrition-induced hyperglycemia and overfeeding, still need to be considered.

In this systematic review and meta-analysis, we examined energy and protein delivery. High energy and protein delivery appeared to prevent physical impairments, whereas only protein delivery attenuated muscle loss and slightly improved ADL. Protein delivery may need to be secured in the acute phase because energy delivery is suppressed to achieve underfeeding. To clarify the effects of energy and protein delivery, further RCTs that focus on physical impairments with more consensus-obtained timing and outcomes are warranted. Furthermore, most studies had an insufficient research structure and limited level of evidence. Therefore, well-structured RCTs on the effects of energy or protein delivery on physical impairments are needed.

### Limitations

This meta-analysis has several limitations. First, only a small number of studies included outcomes, particularly those concerning ADL, physical function, and QOL, although these outcomes are attracting increasing attention as appropriate outcomes for critically ill patients [63]. Some studies were conducted by the same group [26,30,33,34,37,43,53,55]. Second, the observation period of patients was different among studies. Third, we did not separate enteral and parenteral delivery routes due to the small number of studies included. However, parenteral nutritional therapy is not considered to be inferior to enteral nutrition for this effect [4]. Fourth, we focused on nutrition therapy, and the detail of accompanied rehabilitation is unknown especially about the intervention timing and frequency. Fifth, we used two different muscle mass evaluation methods: ultrasound and computed tomography. However, these two evaluations are closely related and reliable [64]. Sixth, there are scarce data about the precision of outcome evaluation methods.

## 5. Conclusions

The current meta-analysis revealed that high protein delivery >1.0 g/kg/day during the first 4 to 10 days of ICU was associated with attenuated muscle loss and slightly improved ADL, while high energy delivery did not exert significant effects. Few RCTs have investigated the effects of energy and protein delivery on more functional outcomes; therefore, future trials are warranted.

## Figures and Tables

**Figure 1 nutrients-14-04849-f001:**
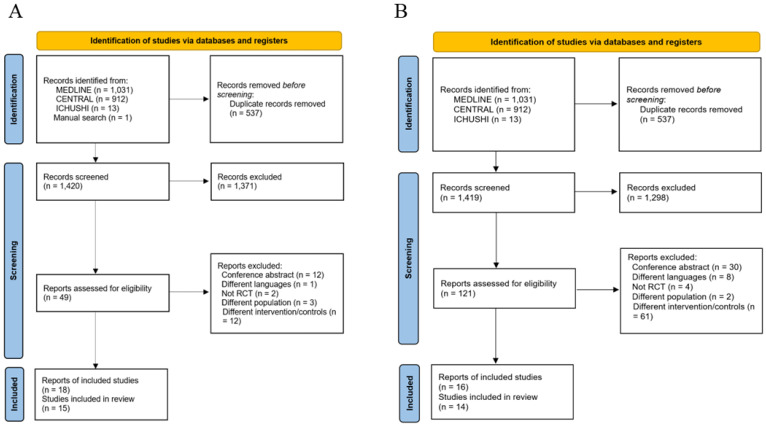
Flow chart of the study included in this systematic review. (**A**) Energy delivery. (**B**) Protein delivery.

**Figure 2 nutrients-14-04849-f002:**
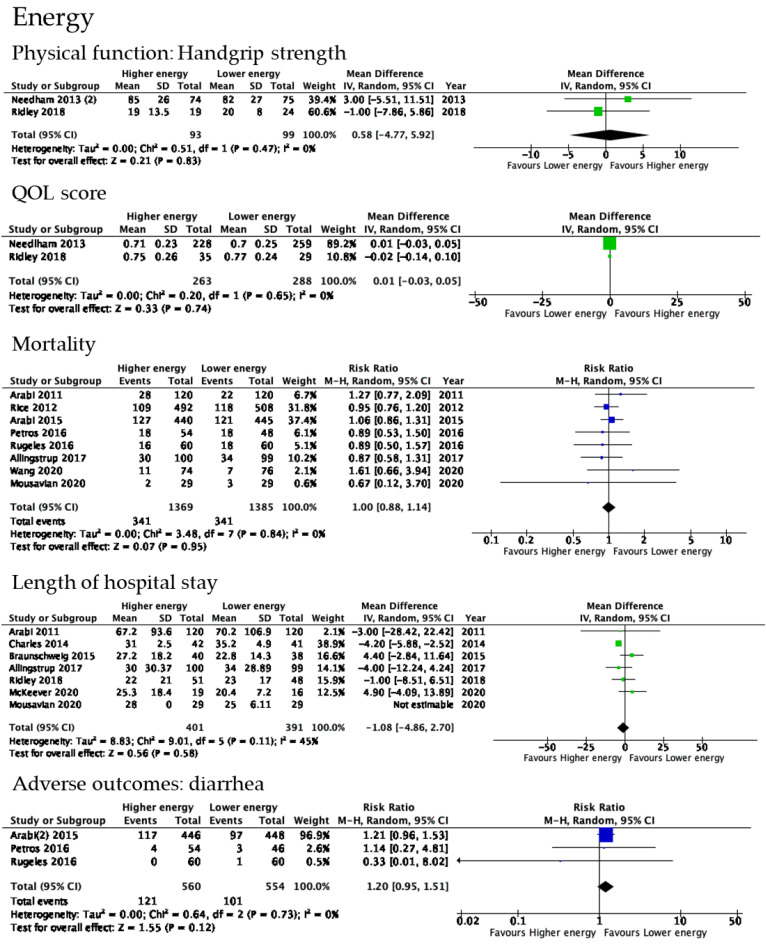
Forest plot of studies included in the optimal energy delivery analysis. Included studies were [26,27] in physical function, [26,27] in QOL score, [29,30,32,34,35,37,39,40] mortality, [28,34], [27,31,39,40,42] in length of hospital stay, [29,32,37] in adverse outcomes.

**Figure 3 nutrients-14-04849-f003:**
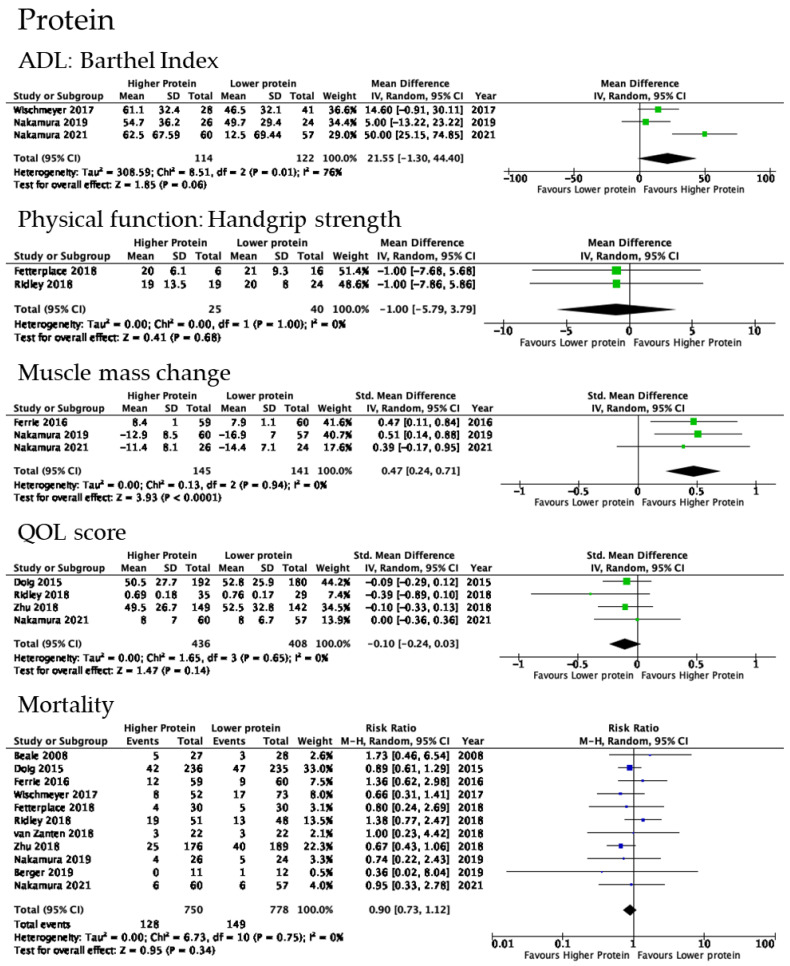
Forest plot of studies included in the optimal protein delivery analysis. Included studies were [52,53,55] in ADL, [27,49] in physical function, [45,53,55] in muscle mass change, [27,53,54,56] in QOL score, [27,44,45,46,48,49,52,53,54,55,56] in mortality, [27,44,45,46,47,48,49,51,52,53,54,55,56] in length of hospital stay, [44,47,48,49,50,53,55] in adverse outcomes.

**Table 1 nutrients-14-04849-t001:** Characteristics of included studies for optimal energy requirements.

Source	Population	No. of Patients			Age (Years)		Intervention			
		Total	High	Low	High	Low	Route	Period	High	Low
Needham 2013 (1) [26]	ALI < 48 h, MV < 72 h, BMI ≥ 25, EN ≤ 48 h > 5 d	487	228	259	52 ± 15	52 ± 16	EN	Assignment up to 6 d	80% of the caloric goal	25% of caloric goal
Ridley 2018 [27]	ICU ≤ 48–72 h, MV, Organ system failure ≥ 1	99	51	48	59 ± 17	60 ± 17	PN	Assignment up to 7 d	100% of estimated energy requirement	Usual clinical practice
Charles 2014 [28]	Surgical ICU, Expected ICU stay > 48 h	83	42	41	53.4 ± 2.7	50.4 ± 2.8	EN+PN	During the ICU stay (10–13 d)	25–30 kcal/kg/day	50% of 25–30 kcal/kg/day
Rugeles 2016 [29]	Expected EN > 96 h	120	60	60	51.8 ± 20.3	53.8 ± 19.0	EN	Assignment up to 7 d	25 kcal/kg/day	15 kcal/kg/day
Rice 2011 [30]	MV ≥ 72 h	200	102	98	54 ± 17	53 ± 19	EN	Initial 6 d of MV	25–30 kcal/kg/day (1418 ± 686 kcal/day in results)	10 mL/h (300 ± 149 kcal/day in results)
Braunschweig 2015 [31]	Mixed ICU, ALI	78	40	38	52.5 ± 17.1	58.6 ± 16.2	EN	Within 24 h of ALI diagnosis to hospital discharge	Indirect calorimetry or 30 kcal/kg	Usual clinical practice
Petros 2016 [32]	Nutritional support ≥ 3 d	100	54	46	64.3 ± 11.5	67.6 ± 11.5	EN+PN	Within 24 h of ICU admission up to 7 d	100% of daily energy expenditure (19.7 ± 5.7 kcal/kg/day in results)	50% of daily energy expenditure (11.3 ± 3.1 kcal/kg/day in results)
Needham 2013 (2) [33]	ALI < 48 h, MV < 72 h	149	74	75	47 ± 14	48 ± 14	EN	Assignment until discharge from the ICU	1300 kcal/day	400 kcal/day
Arabi 2011 [34]	Mixed ICU, Expected ICU stay > 48 h, Glucose > 110 mg/dL	240	120	120	51.9 ± 22.1	50.3 ± 21.3	EN	Within 48 h of ICU admission until discharge from the ICU	90–100% of the Harris-Benedict equation	60–70% of the Harris-Benedict equation
Wang 2020 [35]	Medical ICU, MV, Expected ICU stay > 72 h	150	74	76	57.1–72.3	58.8 ± 70.2	EN+PN	Assignment up to 6 d	25 kcal/kg/day	600 kcal/day
Arabi 2015 [37]	EN < 48 h after ICU admission, Expected ICU stay > 72 h	894	446	448	50.9 ± 19.4	50.2 ± 19.5	EN	Assignment up to 14 d	70–100% of calculated caloric requirements	40–60% of calculated caloric requirements
Mousavian 2020 [39]	Expected ICU stay > 96 h, GCS ≥ 4, ≤10, BMI > 18.5 kg/m^2^	58	29	29	40 ± 16	42 ± 14	EN	Assignment up to 14 d	75% of estimated energy requirement	30% of estimated energy requirement
Allingstrup 2017 [40]	ICU ≤ 24 h, Expected ICU stay > 3 d, BMI > 17 kg/m^2^	199	100	99	63 (51–72)	68 (52–75)	EN+PN	Within 24 h of ICU admission to the ICU discharge	100% of caloric requirements	25 kcal/kg/day
McKeever 2020 [42]	Mixed ICU, Expected MV > 72 h, SIRS	35	19	16	55.6 ± 15.1	57.0 ± 16.6	EN+PN	Assignment up to 7 d	100% of estimated energy requirement (25–30 kcal/kg/day)	40% of estimated energy requirement (10–12 kcal/kg/day)
Rice 2012 [43]	ALI < 48 h, MV ≥ 72 h	1000	492	508	52 ± 16	52 ± 17	EN	Initial 6 d of MV	25–30 kcal/kg/day (1300 kcal/day in results)	10–20 kcal/h (400 kcal/day in results)

ALI: acute lung injury, MV: mechanical ventilation, BMI: body mass index, EN: enteral nutrition, ICU: intensive care unit, GCS: Glasgow coma scale, PN: parenteral nutrition

**Table 2 nutrients-14-04849-t002:** Characteristics of included studies for optimal protein requirements.

Source	Population	No. of Patients			Age (Years)		Intervention			
		Total	High	Low	High	Low	Route	Period	High	Low
van Zanten 2018 [44]	Mixed ICU, MV, BMI ≥ 25, EN ≤ 48 h > 5 d	44	22	22	63.9 ± 13.3	60.8 ± 15.2	EN	From day 1–2 of ICU admission to ICU discharge (up to 28 d)	1.49 g/kg at day 5 in results	0.76 g/kg on day 5 in results
Ferrie 2016 [45]	Mixed ICU, MV within 48 h ≥ 72 h	119	59	60	67.0 (55.5–74.3)	64.5 (49.3–70.0)	PN	From day 1–2 of ICU admission to 10 d	1.2 g/kg (1.1 g/kg in results)	0.8 g/kg (0.9 g/kg in results)
Ridley 2018 [27]	Mixed ICU, MV, BMI ≥ 25, EN ≤ 48 h > 5 d	99	51	48	59 ± 17	60 ± 17	PN	From day 2–3 of ICU admission to 7 d	Supplemental PN	Usual clinical practice
Berger 2019 [46]	MV, Expected ICU stay ≥ 5 d more	23	11	12	63.0 (55.0–73.0)	67.5 (62.3–75.0)	EN+PN	From day 3 of ICU admission	Indirect calorimetry 100% target (1.11 g/kg in results)	Usual clinical practice (0.69 g/kg in results)
Jakob 2017 [47]	Mixed ICU, expected ICU stay ≥ 7 d	90	46	44	65.3 (52.6–75.3)	61.6 (48.6–71.3)	EN	From day 1–3 of ICU admission to 10 d	Caloric target of 25 kcal/kg/day on the third day after enteral nutrition (1.13 g/kg/day)	Caloric target of 25 kcal/kg/day on the third day after enteral nutrition (0.80 g/kg/day)
Beale 2008 [48]	Infection, APACHE II > 10, Expected ICU stay > 5 d, EN > 5 d	55	27	28	57.4 ± 19.0	64.3 ± 16.8	EN	Within 24 h after the enrollment to 10 d	1.4 g/kg/day	0.5 g/kg/day
Fetterplace 2018 [49]	Mixed ICU, MV within 48 h, expected > 72 h	60	30	30	55 ± 13	57 ± 16	EN	From day 1–2 of ICU admission to ICU discharge (up to15 d)	1.2 g/kg/day over the study period	0.75 g/kg/day over the study period
Tuncay 2018 [50]	Neurocritical ICU	46	23	23	73.9 ± 15.3	71.8 ± 20.0	EN	During ICU stay	1.02 g/kg/day at 21 d in results	0.85 g/kg/day at 21 d in results
Reilly 1990 [51]	Patients to undergo liver transplantation	20	10	10	44–50	51 ± 9	PN	Immediately after liver transplantation to 7 d	1.5 g/kg/day	No nutritional support
Wischmeyer 2017 [52]	Mixed ICU, MV, Acute respiratory failure, EN ≤ 48 h, BMI < 25 kg/m^2^, >35 kg/m^2^	125	52	73	55.8 ± 19.8	55.1 ± 16.2	PN	From day 1–2 of ICU admission to 7 d	PN solution (100% calorie goal) (106 g in results)	A standard polymeric solution (100 g in results)
Nakamura 2021 [53]	Mixed ICU, No lower limb injury, No expected death or discharge from the ICU	117	60	57	68.3 ± 14.3	67.9 ± 14.9	EN	From day 1–2 of ICU admission to 10 d	1.8 g/kg/day	0.9 g/kg/day
Doig 2015 [54]	Mixed ICU, Expected ICU stay ≥ 2 d	474	239	235	63.3 ± 15.4	62.7 ± 16.6	PN	From day 1–2 of ICU admission to ICU discharge	100 g of amino acids or maximum 2.0 g/kg/day	Usual clinical practice
Nakamura 2019 [55]	Mixed ICU, No lower limb event, Early expected discharge from the ICU	50	26	24	71.8 ± 12.4	76.6 ± 12.3	EN	From day 1–2 of ICU admission to 10 d	HMB (1.06 g/kg/day at day 7 in results)	Usual clinical practice (0.87 g/kg/day on day 7 in results)
Zhu 2018 [56]	Mixed ICU, Expected ICU stay ≥ 2 d	368	179	189	62.4 ± 15.8	62.3 ± 17.1	PN	From day 1–2 of ICU admission to ICU discharge	2.0 g/kg/day	Usual clinical practice

ICU: intensive care unit, MV: mechanical ventilation, BMI: body mass index, EN: enteral nutrition, APACHEII: acute physiology and chronic health evaluation II, PN: parenteral nutrition, SPN: supplemental parenteral nutrition, HMB: 3-hydroxy-3-methylbutanoic acid

## Data Availability

All reported data are available in the manuscript.

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
