# Peer review of "Impact of Energy and Protein Delivery to Critically Ill Patients: A Systematic Review and Meta-Analysis of Randomized Controlled Trials"

_nutrients, 2022, doi:10.3390/nu14224849_

Round 1

Reviewer 1 Report

Reviewer Comments,

In this article, “Impact of energy and protein delivery to critically ill patients: A systematic review and meta-analysis of randomized controlled trials”. The structure of the manuscript is complete, and the data is substantial.

1. The identification results of the protein delivery should be integrated into the abstract;

2. The keywords are inappropriate;

3. Materials and methods are numbered in confusion.

4. There is no space between some numbers and units.

5. References, the format of the topic of the paper is not uniform.

Reviewer 2 Report

Protein and energy supply in vitro is very important for life support in critically ill patients. Previous studies have not reported much, and this article can quantify energy transport and protein supply well, which has good scientific value and can provide a basis for nutrition supply in critically ill patients.

Author Response

Protein and energy supply in vitro is very important for life support in critically ill patients. Previous studies have not reported much, and this article can quantify energy transport and protein supply well, which has good scientific value and can provide a basis for nutrition supply in critically ill patients.

A. We appreciate reviwer's positive comments. We were so encouraged by your comments. Thanks.

Reviewer 3 Report

The ms may have value after some suitable revisions. My major concerns are as follows,

first, in the method section the authors mention observation periods of up to 1 year. However, most studies shown in the table 1 had observation periods not exceeding the clinical stay of the patients. In any case, the observed changes have to be adjusted for the duration of the intervention, total energy-/protein intake given during the treatment periods as well as the duration of the hospital stay;

second, the authors mix up quantitative (e.g., on muscle mass) and qualitative data (e.g., on ADL). This is confusing because of differences in data quality. It is thus mandatory to characterize the precision of all methods used in these studies. The authors are asked to compare the precision of the methods with the observed and calculated effect sizes;

third, the authors missed to describe details on the methods used, e.g., to assess skeletal muscle mass (SMM). SMM can be measured by whole body CT as well as by taking an individual slice (e.g., at L3) as ‚pars pro totum‘. In that case, it is worthwhile to mention that measurements at one side reflect whole body SMM while they cannot be used to assess changes in SMM. This is because different muscle tissues are affected differently by weight changes. To get the point, the authors are referred to the work of Lisa Schweitzer et al. published in AJCN in 2015. A further point to mention here is that to assess SMM the authors cannot mix up CT and US-data. This is also because there is no widely accepted SOP for the measurement of SMM by US. Thgus, the authors should refer to SMM measured by CT only. Again the precision of CT measurements of SMM should be compared with the observed effect sizes;

fourth, going through the individual plots it becomes evident that for individual outcomes (e.g., for SMM) there are only three studies included in the meta-analysis. Obviously, that low no of studies are not a sufficient basis to perform that kind of analysis. In the case of the outcome SMM it is to mention that 2 out of 3 studies have been done by the same group of authors which again questions that analysis;

fifth, the authors have mentioned ADL (measured by the Barthel Index) as their primary outcome. However, following their data presentation is rather confusing mixing up the primary outcome with secondary outcomes. There is need of clear structuring of the results and discussion section. Since I have the impression that changes in ADL and changes in other outcomes have been determined at different times within or after the hospital stay this has to be documented in detail;

sixth, since energy and nitrogen had been supplied together the energy to nitrogen ratio has to be taken into account too. As far as the initial (or post hoc?) calculation of energy and nitrogen needs of the patients assessed  in the different studies is concerned there is need to provide details of the algorithms used for that calculations. Since different studies have been included it is likely that using different calculations will impact the categorization used in this meta-analysis. In any case there is need of some re-calculations of the data.

Round 2

Reviewer 3 Report

The authors have improved their ms. However, my point still is about the precision of the methods applied. The authors response suggests that precision of the methods is unknown. This is not true. E.g., there are data on the precision of the assessment of SMM by single slice CT. This is also true for estimating SMM by US. It is a matter of search strategy to find that data. The authors should accept that without knowing the precision of methods the effect sizes cannot be judged.
